# Gender-Specificity of Fatigue and Concerns Related to the COVID-19 Pandemic—A Report on the Polish Population

**DOI:** 10.3390/ijerph20075407

**Published:** 2023-04-05

**Authors:** Katarzyna Domosławska-Żylińska, Magdalena Krysińska-Pisarek, Dorota Włodarczyk

**Affiliations:** 1Department of Education and Communication, National Institute of Public Health NIH—National Research Institute, Chocimska 24, 00-791 Warsaw, Poland; 2Department of Heath Psychology, Medical University of Warsaw, ul. Litewska 14/16, 00-575 Warsaw, Poland

**Keywords:** pandemic fatigue, women’s health, COVID-19, concerns

## Abstract

Background: The COVID-19 pandemic, which is now going on its third year, and its consequences experienced in almost every sphere of life may eventually lead to pandemic fatigue. Previous research indicates that one of the important determinants of the negative consequences of the pandemic is gender. Based on the WHO’s definition of pandemic fatigue, a study was undertaken to determine the level of pandemic fatigue and concerns expressed in relation to the pandemic in Poland. Methods: A survey among 1064 respondents was conducted using the CATI technique during the fourth wave of the COVID-19 pandemic in Poland. Questionnaires adopted: the Polish adaptation of the Pandemic Fatigue Scale (PFS), Subjective Fatigue Symptoms (SFS), and the COVID-19 Concerns Checklist (CCC) and sociodemographic questions. The following statistical methods were employed: ANOVA analysis of variance, Mann–Whitney U test or Kruskal–Wallis test, and Chi-square test. Results: The analysis showed that women received higher overall PFS, information PFS, SFS, and pandemic-related concerns scores. Other factors that were associated with pandemic fatigue were age, treatment for chronic diseases, receipt of the COVID-19 vaccine, and COVID-19 survivor status. Conclusions: Women in Poland are experiencing pandemic fatigue, subjective fatigue symptoms, and concerns associated with the COVID-19 pandemic at a higher rate than men. Along with implementing interventions focused on public health safety, it seems reasonable to put in place strategies to assist people who are less capable of coping with prolonged stressful situations.

## 1. Introduction

The COVID-19 pandemic, which is now in its third year, does not continue without leaving a scar on the social, psychological, or economic aspects of people’s lives [1,2]. The sinusoidal waveform of the pandemic, associated with alternating increases and decreases in the number of infections and the tightening and loosening of restrictions on preventive behaviour, requires a constant (with periods of higher and lower) commitment from the public to reduce the impact of the pandemic [3]. The emergence of new COVID-19 variants, such as Delta and Omicron, prolonged the pandemic, resulting in, among other things, a gradual habituation to the risks of the virus and a reduced commitment to the use of non-pharmaceutical interventions [4]. The COVID-19 pandemic has been accompanied by an abundance of information from different sources about the coronavirus, and the frequent contradictions lead to confusion and information overload for the public [5]. Prolonged exposure to multiple and potentially dangerous agents may result in intensified concerns on the one hand and the development of pandemic fatigue on the other.

To this date, several concepts of COVID-19 pandemic fatigue have come to light. It is possible to identify approaches that mainly refer to fatigue resulting from the experience of the pandemic. One example is the WHO concept, which focuses on information and behavioural aspects and defines pandemic fatigue as a decrease in motivation to engage in protective behaviours and to seek COVID-19-related information, which may be accompanied by feelings of loneliness, alienation, and hopelessness [4]. This approach is particularly relevant in the field of public health, as it concentrates on reduced social involvement, which may adversely affect the efforts made to control the spread of the virus and manage the COVID-19 pandemic. Taylor (2021) also describes COVID Disregard Syndrome, which is manifested by behaviours indicating a belief that the risks associated with COVID-19 are exaggerated [6].

A different approach was proposed by Rudroff et al. (2020), which focuses on post-COVID-19 fatigue [7]. He defines it as “the decrease in physical and/or mental performance that results from changes in central, psychological, and/or peripheral factors due to the COVID-19 disease,” influenced mainly by psychophysical factors (stress, anxiety, depression, and pain), inflammations and pre-existing medical conditions (e.g., asthma and hypertension) [7]. This means that such defined fatigue is not so much a result of the experience of the pandemic (and the stress associated with it) but rather of experiencing COVID-19 itself, the effects of comorbidities, or the physiological effects of mental disorders. Regardless of the psychophysical factors mentioned by Rudroff, numerous psychophysical reactions to the COVID-19 pandemic, such as emotional exhaustion, inability to function efficiently, reduced motivation, difficulty sleeping, feelings of helplessness, and hopelessness, are also indicated [8,9,10,11]. Therefore, it seems that extending the information and behavioural approach to psychophysical symptoms to the experience of the pandemic would provide an opportunity for a deeper analysis of this phenomenon. When analyzing fatigue, two aspects of fatigue were considered: information and behavioural fatigue and subjective psychophysical symptoms.

The effects of the COVID-19 pandemic were or are being felt in almost every sphere of life, forcing a change in the lifestyles of individuals. The increased incidence of infections and disease cases naturally raised concerns about one’s own health and the health of one’s loved ones, but also about the further course of the pandemic, one’s ability to function on a daily basis, and the distant professional or economic impact [12]. What needs to be emphasised from a public health perspective is that the level of concern can affect people’s responses to protective behaviour guidelines [13]. This relationship is curvilinear, meaning that both too low and too high levels of concern will not be conducive to adopting or maintaining such behaviours. It is therefore important to understand the fears felt by the public in the context of managing health security during a pandemic [14]. Research to date suggests that factors of concern in the context of a COVID-19 pandemic include the health of loved ones, health system overload, and economic consequences for individuals [15].

Gender is among the factors influencing the level of experiencing negative consequences of the pandemic [16]. Studies have shown that women experience the negative consequences of the pandemic, leading to changes within their daily lives, to a greater extent than men. Women were more likely to take over childcare due to the closure of schools and childcare facilities, combining work duties and household chores. According to the OECD, during times of crisis and quarantine, women are also more vulnerable to economic consequences (loss of work and income) and increased risk of violence, exploitation, abuse, or harassment [16]. Women were also more likely to experience moderate to high levels of anxiety related to COVID-19 [17]. Another group that is significantly affected by the negative effects of the pandemic is young people. In addition to gender and age, cited are such factors as the level of education, job status, and place of residence [18,19].

Given the considerable effects of pandemic fatigue and concern, as well as taking into account the current development of the pandemic and the possibility of further pandemic waves, the monitoring of these phenomena and their further analysis seem essential. The aim of this study is to determine the level of pandemic fatigue and concerns experienced in relation to the pandemic in Poland. Two specific objectives of the study were identified: (1) Determining the role of gender in the level of both aspects of pandemic fatigue and of pandemic concerns; (2) Testing the role of other determinants of the level of both aspects of pandemic fatigue and of pandemic concerns. The study appears to be unique due to the small number of scientific reports on pandemic fatigue and concerns related to COVID-19 from Central and Eastern Europe. The results may be helpful in identifying the steps necessary to build an appropriate response to the next waves of the pandemic.

## 2. Materials and Methods

### 2.1. Recruitment of Participants and Procedure

A random sample representative of the Polish population aged 18 and over with a minimum size of n = 1060, with the following assumptions:(a)The sample size in each voivodeship should be proportional to the population of the voivodeship;(b)In each voivodeship, the proportion of sex and age (18–39; 40–59; 60+) needs to be maintained;(c)In addition, the proportion of urban and rural residents by sex and age (18–39; 40–59; 60+) must be maintained nationwide.

The survey was carried out during the 4th wave (20 November 2021–15 December 2021, daily number of infections: 23–29 thousand) of the pandemic in Poland [3]. A cross-sectional survey among the respondents was conducted using the CATI (Computer Assisted Telephone Interview) technique based on computer-supervised telephone interviews by a professional survey company. The data source for contacting respondents was a sampling operator in the form of a database of contact numbers held by the company conducting the survey. This operator included a database of landline and cell phone numbers operating in Poland. The database used did not include telephone numbers registered by business entities and those included in the so-called Robinson List (a list of telephone numbers of private subscribers under which a categorical refusal to make contact was recorded). Informed consent was obtained from all the subjects involved in the study. Due to the scope of the data, we obtained verbal informed consent. The study employed the CATI method of data collection. Pursuant to applicable regulations in Poland, CATI testing does not require approval from the Ethics Committee.

### 2.2. Questionnaire

The survey questionnaire consisted of four parts. The first part included questions relating to sociodemographic data (age, gender, place of residence, education) and basic health information (whether the respondent had had COVID-19, was being treated for chronic diseases, and whether they had been vaccinated) (Appendix A). The second and third parts of the questionnaire were used to explore the level of pandemic fatigue in both aforementioned aspects. The information and behavioural fatigue were examined using the Polish adaptation of the Pandemic Fatigue Scale (PFS) derived from the COSMO study (COVID-19 Snapshot MOnitoring) (Appendix A). The back translation method was used for accurate linguistic adaptation. The scale contains six statements relating to attitudes towards information and behaviours regarding the COVID-19 pandemic, rated by the respondents on a Likert scale from one to seven, where one means strongly disagree, and seven means strongly agree. Two specific elements can be assessed using the PFS: information fatigue (being tired of hearing about COVID-19, PFS info) and behavioural fatigue (feeling unmotivated and discouraged in the fight against COVID-19, PFS behav), as well as global score (the sum of all items). The Cronbach’s alpha reliability coefficient for the study group is 0.91 (PFS info) and 0.88 (PFS behav). Psychophysical symptoms of pandemic fatigue were investigated using the Subjective Fatigue Symptoms scale-6; SFS-6) (Appendix A). It includes the following 4 reactions: cognitive, emotional, sensory, and somatic. The cognitive ones encompass difficulty concentrating and organising one’s own mental activity; emotional ones include emotional tension and/or mental discomfort. Sensory ones are related to feeling overwhelmed by the need to stay in touch with people over the Internet only while feeling a lack of real human contact, feeling an excess of stimuli related to the use of technology for online contact (e.g., the abundance of applications, the frequency of online meetings, the use of the camera). Somatic reactions include a sense of lack of energy and/or reduced capacity to perform daily tasks and sleep problems—sleepiness despite adequate sleep duration or insomnia, disrupted sleep cycle. The respondents indicated the extent to which they felt the above aspects of pandemic fatigue using a 5-point scale, where one meant “none”, and five meant “very intense”. The score is the average of the responses, where a higher score indicates greater intensity of fatigue symptoms. The Cronbach’s alpha reliability coefficient for the study group is 0.89. The fourth part of the questionnaire is the COVID-19 Concerns Checklist (CCC) (Appendix A), where a 5-point scale (with 1 meaning “none” and 5 meaning “very intense”) was used to examine the extent to which respondents relate to the 10 impediments or risks associated with the COVID-19 pandemic. The score is the mean of the obtained responses, where a higher score indicates a greater level of concern about the COVID-19 pandemic. The Cronbach’s alpha reliability coefficient for the study group is 0.86.

### 2.3. Statistical Analyses

Compliance with the normal distribution of quantitative variables was verified using the Kolmogorov–Smirnov test. ANOVA analysis of variance was used for intergroup comparisons relating to aggregate indicators (with Scheffe or Tamhane tests for correction in multiple comparisons, depending on the homogeneity of variance) and non-parametric Mann–Whitney U tests (for two groups) or Kruskal–Wallis tests (for three or more groups) were used for significant deviations from the normal distribution. In order to keep the tables consistent, the results for aggregate indicators are presented as M and SD (in all cases, parametric and non-parametric test results were consistent). A Chi-square test was used to compare frequency distributions (nominal and ordinal scales).

In order to check the internal consistency of the scales used, their reliability was analysed using Cronbach’s alpha reliability coefficient. A value of 0.05 was used as the significance level. All calculations were performed in the R1 (version 4.0.0) and SPSS Statistics 28 software.

## 3. Results

### 3.1. Subjects

The characteristics of the group are presented in Table 1. There were 1060 respondents included in the study. People between 18 and 39 years old accounted for 35.5% of the respondents, 33.3% of the respondents were between 40 and 59 years old, while respondents over 60 years old constituted 31.1% of the group. The largest number of respondents lived in the countryside, the smallest in cities, between 200,000 and 500,000 inhabitants. The largest group among respondents in terms of education were people with secondary or post-secondary education. Baseline health characteristics indicated that more than half had not been treated for chronic disease, while one in ten respondents had more than one such disease. Subjects fully vaccinated against COVID-19 predominated, but unvaccinated subjects accounted for one-third. Status post COVID-19 was confirmed by 17.9%; 64.8% had never contracted the disease, while the remainder were unable to state this clearly.

Data on the prevalence of individual manifestations of cognitive and behavioural fatigue (PFS) in the whole group are provided in Table 2. Descriptive statistics for the aggregate indicators are as follows: global score M = 4.42 (SD = 1.54), PFS info M = 4.68 (SD = 1.63), PFS behav M = 4.18 (SD = 1.65). Analysing the SFS (Table 3) reported by the Polish population (responses ranging from quite intense to very intense), most (25.3%) of the respondents are overwhelmed by the need to stay online while feeling a lack of human contact, and 23% feel a lack of energy and/or reduced capacity to perform daily tasks. The least respondents feel difficulties in focusing their attention and organising their own mental activity—13.1% of the respondents. Descriptive statistics for the symptom aggregate score are M = 2.52 (SD = 0.96).

Out of the concerns surveyed (CCC; Table 4), the respondents mostly indicated those related to the economic situation (the responses “large extent” and “very large extent” were analysed). Rising prices caused concern in 77.8% of the respondents, while 61.3% of the respondents indicated that they were concerned about the economic crisis. More than half of the respondents were concerned about the overloaded health system (53.4%), the unpredictability and uncertainty of the course of the pandemic (52.5%), and a renewed lockdown (52.2%). The respondents were least concerned about becoming a source of infection for their loved ones (17.7%). Descriptive statistics for the aggregate indicator are M = 3.38 (SD = 0.73).

### 3.2. Pandemic Fatigue and Concern by Gender—Comparison of Aggregate Indicators

Statistical analysis showed differences between men and women in the values of the aggregate indicators: overall pandemic fatigue (PFS), information fatigue (PFS info), subjective symptoms of pandemic fatigue (SFS), and pandemic concern (CCC).

Considering aggregate indicators, there were significant differences between women and men in terms of pandemic fatigue (F = 7.78; *p* = 0.005), specifically information fatigue (F = 11.70; *p* < 0.001) and subjective fatigue symptoms (F = 8.56; *p* = 0.003). Women had higher global pandemic fatigue (M = 4.55; SD = 1.51 vs. M = 4.29; SD = 1.56, respectively), information fatigue ((M = 4.84; SD = 1.59 vs. M = 4.50; SD = 1.67, respectively), and subjective symptoms (M = 2.60; SD = 0.93 vs. M = 2.43; SD = 0.97, respectively) than men. There was no difference in behavioural fatigue (F = 3.34; *p* = 0.068). There was also a difference in the global score of pandemic concerns (F = 29.83; *p* < 0.001). Women had a higher level of concerns than men (M = 3.50; SD = 0.70 vs. M = 3.25; SD = 0.74, respectively).

### 3.3. Pandemic Fatigue and Gender—Analysis of Specific Indicators According to the Pandemic Fatigue Scale (PFS)

Analysis of individual statements showed that women were statistically more likely than men to agree with the statement that they were tired of hearing discussions about COVID-19, to admit that they try to change the subject when friends or family talk about COVID-19 because they no longer want to talk about it and to feel overwhelmed by following all the COVID-19-related recommendations (Table 2).

### 3.4. Subjective Symptoms of Pandemic Fatigue and Gender—Analysis of Specific Indicators According to the Subjective Fatigue Symptoms Scale (SFS-6)

Women were more likely than men to feel overwhelmed by the need to stay in touch with people over the Internet only and the lack of real human contact, to feel over-stimulated by the use of technology for online contact, to experience sleep problems and to feel lack of energy and/or reduced capacity to perform daily tasks (Table 3).

### 3.5. COVID-19-Pandemic-Related Concerns by Gender—Analysis of Specific Indicators According to the COVID-19 Concerns Checklist

Women demonstrated stronger concerns about COVID-19 than men. This was true for most of the concerns analysed, with the exception of the concern about being a source of infection for loved ones (Table 4).

### 3.6. Other Determinants of Fatigue and Concern about the COVID-19 Pandemic 

Other factors that were associated with pandemic fatigue were age, being treated for chronic diseases, receiving the COVID-19 vaccine, and having COVID-19 survivor status.

Older people (≥60 years) were considerably less likely to experience COVID-19 pandemic fatigue (F = 33.114, *p* < 0.001), including information fatigue (F = 28.184, *p* < 0.001)) and behavioural fatigue F = 29.572, *p* < 0.001), especially when compared to those in the 30–39 age range. Additionally, the 60+ age group was less likely to experience subjective symptoms of pandemic fatigue (F = 13.526, *p* < 0.001) compared to younger respondents (18–29 and 30–39 age group) (Appendix A).

Subjects not treated for chronic diseases showed greater pandemic fatigue in terms of PFS global (F = 11.131, *p* < 0.001), PFS info (F = 14.201, *p* < 0.001), and PFS behav (F = 6.263, *p* = 0.002) compared to those treated for more than one chronic disease. There were no statistically significant differences in terms of SFS and CCC (Appendix A).

Unvaccinated persons (approx. 30%) demonstrated higher levels of pandemic fatigue (F = 66.859, *p* < 0.001), information fatigue (F = 60.555, *p* < 0.001), and behavioural fatigue (F = 56.890, *p* < 0.001) compared to fully vaccinated persons. People vaccinated with one dose (approximately 5%) experienced subjective symptoms of pandemic fatigue (SFS) to a greater extent compared to those fully vaccinated (F = 7.343, *p* < 0.001). In contrast, those who had received full vaccination felt more anxious compared to those who were not vaccinated (F = 6.127, *p* = 0.002) (Appendix A).

Respondents who were unable to determine whether they had had COVID-19 felt greater pandemic fatigue (PFS global and PFS info) compared to those who had or had not suffered from COVID-19 (F = 5.709, *p* = 0.003 and F = 6.512, *p* = 0.002, respectively). In contrast, people who had had COVID-19 or did not know whether they had had COVID-19 felt subjective aspects of pandemic fatigue to a greater extent than those who believed that they had not been ill (F = 10.171, *p* < 0.001). COVID-19 survivor status did not differentiate the level of concern (Appendix A).

## 4. Discussion

The aim of this study was to analyse pandemic fatigue and concerns. An expanded view of fatigue was used, including its two aspects: information and behavioural aspect, as well as subjective psychophysical symptoms. The determinants of pandemic fatigue and concerns were examined in a sample representative of the Polish population, with a particular focus on gender. The study showed that women had significantly higher scores on pandemic fatigue (especially the information aspect), perceived subjective symptoms of the pandemic, and concerns. The information fatigue is mainly manifested by an aversion to participating in conversations about COVID-19, listening to discussions held by others, and feeling overwhelmed by the regime of protective guidelines of behaviour. High exposure to messages leads to information fatigue and may result in lower adherence to precautions such as hand washing and social distancing [20]. A study on pandemic fatigue conducted in Germany and Denmark found that pandemic fatigue was significantly associated with a reduction in the frequency of protective behaviours: social distancing, hygiene practices, mask wearing, and information seeking [21]. Interestingly, in the group we studied, there were no gender differences in terms of behavioural fatigue, i.e., willingness to comply with protective recommendations or motivation to fight COVID-19. This is an important result confirming that each of these two aspects of fatigue may have its own gender-dependent specificity and possibly other health and social consequences as well.

In addition to COVID-19-related information fatigue, women had higher scores for being overwhelmed by the imbalance between online and real-life interactions with people, sensory over-stimulation with online technology, sleep problems, and lack of energy to carry out daily tasks than men. It is worth noting that no differences were found in the emotional (mental discomfort) and cognitive (attention and mental activity) spheres. The results of the study on the long-term effects of the pandemic indicate that women are more likely to report symptoms such as anxiety, general fatigue, and muscle pain [22,23]. There is also a higher risk of mental disorders and loneliness among women [24]. Again, our study found that psychosocial symptoms of fatigue at any given time may be heterogeneous and demonstrate gender-related variability. This raises the question of to what extent the pandemic generates these symptoms and to what extent it is an exacerbating factor for already existing problems in women (e.g., sleep problems and lack of energy to carry out daily tasks). It is known, for example, that the incidence of depression in women is higher than in men [25].

Referring to the obtained results of contemporary concepts of gender differences, it is worth pointing out two of them. The concept of gender differences in emotionality says, among others, that women have greater sensitivity to emotional stimuli, which is accompanied by experiencing anxiety, less control, and a more negative perception of the experienced burden [26] In turn, the concept of gender roles would suggest that obtained differences between women and men in pandemic fatigue and concerns may be rather related to socio-cultural patterns and expectations [27] (Femininity is generally oriented towards interpersonal contacts and is associated with caring, expressiveness, and sensitivity. Men are characterised by orientation to task, action, independence, and dominance. Thus, the interaction between gender roles and the specific situational demands of the pandemic might be meaningful here. At the same time, no differences in emotional tension and psychological discomfort were found, which would suggest that emotionality would not be crucial here.

The pandemic may have created conditions that are socially and psychologically more difficult for women than for men—women’s family life and career are more based on interpersonal relationships, while the use of information technology is more typical of men’s daily and professional life. These questions require further research. They also raise the issues of definition and appropriate choice of criteria differentiating between different aspects of COVID-19-pandemic-related fatigue. The “two-component” approach employed in this study draws on both the WHO proposal and a broader clinical approach [4,7]. It has made it possible to show the diverse nature of pandemic fatigue and its gender specificity, which is an added value of this work.

According to the results obtained, women experience greater concerns than men with regard to all the risks analysed, with the exception of being a source of infection for loved ones. The economic and financial consequences (e.g., price increases and risk of economic crisis) constitute the greatest concern, which is stronger than the medical ones (e.g., health care system overload, the unpredictability of the course of the pandemic, or renewed lockdown). Increased financial fears and increased childcare duties among women are confirmed by US research [28]. The results obtained suggest that, from the point of view of the economy, the COVID-19 pandemic will affect women more than men. Reasons cited include women’s lower earnings and lower savings, the increased burden of unpaid care for loved ones and housework (which reduces hours in the labour market), and the fact that the majority of single-parent households are women [29]. It is known from other studies that women have higher levels of anxiety than men, which may indicate their greater vulnerability and suggest that greater concerns among women may be a reflection of a general trend [30].

A question arises about the consequences of higher levels of pandemic concerns in women. On the one hand, higher levels of anxiety are conducive to coping with stress focused on emotion rather than the problem [31]. Such coping mechanisms consist mainly of worrying, grieving, ruminating, and trying to release emotions. In excess, they tend to be destructive and can contribute to impaired functioning, e.g., in the form of fatigue, and to the development of mental problems in the long term. A certain dose of worry can promote awareness of the situation and problem solving. If it also helps to manage one’s emotions, it can be very beneficial when external circumstances are difficult to change or control, as was the case with the COVID-19 pandemic [32]. The above assumptions may indicate directions for further research. This study clearly shows that women are a special risk group for pandemic fatigue and its post-pandemic consequences.

Other groups at risk of pandemic fatigue and its consequences are young people (18–39 years old), people who are not treated for chronic diseases, unvaccinated people, and people who have had COVID-19. The COVID-19 pandemic affected young adults in many areas, including productivity (work, study, and parenting), social relationships (meetings with friends, concerts, and parties), and leisure (use of fitness clubs and travel). Subjects who were students additionally mentioned the impact on education, while those who had completed their education focused on the imbalance between work and family life and insecure financial situation. Early adulthood is associated with making decisions that shape the future in terms of career and social relationships. Therefore, the COVID-19-pandemic-related situation and uncertainty about the future may result in young adults experiencing increased levels of negative consequences of the pandemic. According to the study, young adults, compared to older age groups, reported greater problems with pandemic fatigue. The findings are in line with other studies, where young adults reported the highest levels of stress, the least ability to cope with this stress, and the greatest difficulty in making daily and important decisions [32]. They were more likely to report such symptoms as headaches, fatigue, and sleep issues [32,33]. In the context of young and working people, a question arises as to what extent pandemic fatigue may contribute to, exacerbate or accelerate the process of professional burnout [34]. It is known that one of the first symptoms of this syndrome is psychophysical exhaustion, which can also be manifested by fatigue and difficulty in recovering one’s strength. These phenomena can be difficult to differentiate.

Higher levels of pandemic fatigue among unvaccinated people may be associated with the attitude displayed by anti-vaccination groups towards the pandemic. Those who disagree with the measures adopted to prevent the pandemic (e.g., vaccines) are also characterised by their low COVID-19 risk assessment (no differences in the CCC area), lack of trust in authorities, high uncertainty in distinguishing between true and false messages, and high interest in social media. For those exhibiting a high belief in the COVID-19 conspiracy and denial of the pandemic, any restrictions or safety rules are not necessary, and adapting to them results in higher levels of fatigue [35,36].

It is notable that COVID-19 survivor status is associated with increased levels of experiencing fatigue, particularly in terms of subjective fatigue symptoms (SFS). This result is congruent with Rudroff’s study and concept, which analysed the determinants and correlates of post-COVID fatigue. This result points to the need for further research to be able to specify and differentiate the different aspects of fatigue.

The results obtained constitute an important addition to existing knowledge. The survey included a representative sample of the Polish population; the research tools showed high reliability while conducting the interviews by means of telephone interview allowed for reaching a wider group of respondents than in the case of an online survey.

Limitations of the survey include the fact that the survey is cross-sectional and provides information only from a certain time of the pandemic (4th wave), without being able to monitor the dynamics of fatigue and concerns. The data obtained from respondents are declarative. The survey questionnaire did not contain questions about mental problems occurring before the pandemic. It can be difficult to compare the results obtained with other studies, as the specificity of the pandemic situation is crucial, and obtaining precise characteristics is not always possible (e.g., number of cases, number of deaths, nature of the lockdown (e.g., China vs. Europe), its duration or the health policy of the given country). Most studies do not include such characteristics, and the levels of fatigue and concern may be associated with them. This study focused primarily on the role of gender and only selected other differentiating factors. When undertaking similar studies on a larger population, consideration may be given to additionally performing multivariate analysis, allowing for differential results, particularly between people ≥60 years old and people treated for chronic diseases. In future studies, it would be worth considering other systemic and individual factors as well as broadening the analyses to include multivariate analyses and a search for interaction effects. The study suggests that gender may moderate the relationship between some factors and fatigue and concerns. This is considered the next stage of this project.

Ours, and other studies, show that women experience the long-term effects of the COVID-19 pandemic, including pandemic fatigue, to a greater extent than men. It is essential to take into account pandemic fatigue and the long-term psychophysical effects of the COVID-19 pandemic when planning measures to reduce the spread of the virus and ensure public health safety. According to WHO recommendations, under pandemic conditions, protective measures should be sought that will to the lowest extent, disrupt the current functioning and overexert human resources. It is also advisable to identify, as soon as possible, those individuals who are more vulnerable and/or less able to cope with prolonged stressful situations and to implement appropriate strategies to support these individuals [36].

## 5. Conclusions

Women in Poland are experiencing pandemic fatigue, subjective fatigue symptoms, and concerns associated with the COVID-19 pandemic at a higher rate than men. Other groups vulnerable to pandemic fatigue and its consequences include young people (18–39 years old), people who are not being treated for chronic diseases, unvaccinated people, and people who have suffered from COVID-19. Along with implementing interventions focused on public health safety, it seems reasonable to put in place strategies to assist people who are less capable of coping with prolonged stressful situations.

## Figures and Tables

**Table 1 ijerph-20-05407-t001:** Characteristics of the respondents.

		Women % (n)	Men % (n)	Total % (n)
		52.4 (555)	47.6 (505)	100 (1060)
Age	18–29	15.6 (86)	17.0 (86)	16.2 (172)
	30–39	18.0 (100)	20.8 (105)	19.3 (205)
	40–49	18.7 (104)	24.4 (123)	21.4 (227)
	50–59	13.3 (74)	10.3 (52)	12.0 (126)
	≥60	34.4 (191)	27.5 (139)	31.1 (330)
Accommodation	Countryside	36.8 (204)	39.0 (197)	37.9 (401)
	Town ≤200,000	31.7 (176)	27.5 (139)	29.7 (315)
	Town 200,000–500,000	14.2 (79)	15.5 (78)	14.8 (157)
	Town ≥500	17.3 (96)	18.0 (91)	17.6 (187)
Education	Elementary or junior high school	8.6 (48)	8.7 (44)	8.7 (92)
	Basic vocational	20 (111)	29.9 (151)	24.7 (262)
	Secondary or post-secondary	39.3 (218)	33.1 (167)	36.3 (385)
	Higher education	32.1 (178)	28.3 (143)	30.3 (321)
Employment	Employed (full-time or self-employed)	56.2 (312)	67.7 (342)	61.7 (654)
	Student	2.5 (14)	5.5 (28)	4.0 (42)
	Unemployed	3.6 (20)	3.2 (16)	3.4 (36)
	Pensioner/Retiree	31.7 (176)	22.6 (114)	27.4 (290)
	Household leader	6.0 (33)	1.0 (5)	3.5 (38)
Chronic diseases	No disease under treatment	57.3 (318)	57.2 (289)	57.3 (607)
	1 disease	32.1 (178)	29.9 (151)	31.0 (329)
	More than 1 disease	10.6 (59)	12.9 (65)	11.7 (124)
COVID-19 vaccination	Not taking the vaccine	34.2 (190)	26.3 (133)	30.5 (323)
	Only first dose	3.2 (18)	7.5 (38)	5.3 (56)
	Full vaccination	62.5 (347)	66.1 (334)	64.2 (681)
COVID-19 status	Recoverd	17.1 (95)	18.8 (95)	17.9 (190)
	Have not been ill	66.1 (367)	63.4 (320)	64.8 (687)
	I do not know	16.8 (93)	17.8 (90)	17.3 (183)

**Table 2 ijerph-20-05407-t002:** Individual items of PFS (M, SD, and frequencies) by gender (n = 1060; n (%)).

	M (SD)	1	2	3	4	5	6	7	Chi^2^ (*p*)
I’m sick of hearing about COVID-19									
Women	5.02 (1.75)	22 (4.0)	44 (7.9)	59 (10.6)	52 (9.4)	111 (20.0)	131 (23.6)	136 (24.5)	17.176 (<0.001)
Men	4.65 (1.87)	28 (5.5)	54 (10.7)	78 (15.4)	52 (10.3)	101 (20.0)	80 (15.8)	112 (22.2)	
2. When friends or family members talk about COVID-19, I try to change the subject because I do not want to talk about it anymore.									
Women	4.61 (1.69)	20 (3.6)	62 (11.2)	67 (12.1)	84 (15.1)	134 (24.1)	106 (19.1)	82 (14.8)	22.475 (<0.001)
Men	4.22 (1.73)	34 (6.7)	58 (11.5)	85 (16.8)	101 (20.0)	100 (19.8)	66 (13.1)	61 (12.1)	
3. I feel strained from following all of the behavioural regulations and recommendations around COVID-19									
Women	4.65 (1.74)	23 (4.1)	63 (11.4)	70 (12.6)	67 (12.1)	128 (23.1)	115 (20.7)	89 (16.0)	16.644 (<0.05)
Men	4.31 (1.85)	39 (7.7)	61 (12.1)	85 (16.8)	69 (13.7)	99 (19.6)	74 (14.7)	78 (15.4)	
4. I am tired of all the COVID-19 discussions in TV shows, newspapers, radio programs, etc.									
Women	4.90 (1.75)	20 (3.6)	52 (9.4)	60 (10.8)	59 (10.6)	139 (25.0)	93 (16.8)	132 (23.8)	8.606 (0.197)
Men	4.64 (1.83)	29 (5.7)	49 (9.7)	75 (14.9)	58 (11.5)	111 (22.0)	78 (15.4)	105 (20.8)	
5. I am tired of restraining myself to save those who are most vulnerable to COVID-19.									
Women	4.00 (1.88)	55 (9.9)	92 (16.6)	94 (16.9)	85 (15.3)	85 (15.3)	76 (13.7)	68 (12.3)	5.924 (0.635)
Men	3.94 (1.95)	60 (11.9)	85 (16.8)	80 (15.8)	86 (17.0)	63 (12.5)	56 (11.1)	75 (14.9)	
6. I am losing my spirit to fight against COVID-19.									
Women	4.15 (1.75)	31 (5.6)	89 (16.0)	85 (15.3)	121 (21.8)	84 (15.1)	79 (14.2)	66 (12.0)	9.541 (0.142)
Men	3.99 (1.86)	48 (9.5)	89 (17.6)	67 (13.3)	107 (21.2)	72 (14.3)	55 (10.8)	67 (13.3)	

1—I strongly disagree; 2—I disagree; 3—I rather disagree; 4—No opinion; 5—I rather agree; 6—I agree; 7—I strongly agree.

**Table 3 ijerph-20-05407-t003:** Individual items of SFS-6 (M, SD, and frequencies) by gender (n = 1060; n (%)).

SFS-6		M (SD)	Not at All	To a Small Extent	Moderately	Fairly Intensively	Very Intensively	Chi^2^ (*p*)
Feeling overwhelmed by the need to stay connected online with people while feeling a lack of real human contact	women	2.76 (1.23)	115 (20.7)	113 (20.4)	163 (29.4)	119 (21.4)	45 (8.1)	16.444 (<0.01)
	men	2.50 (1.21)	130 (25.7)	132 (26.1)	138 (27.3)	69 (13.7)	36 (7.1)	
A sense of over-stimulation related to the use of technology for online contact (e.g., multiple apps, frequency of online meetings, use of camera, etc.).	women	2.63 (1.20)	131 (23.6)	113 (20.4)	137 (31.2)	104 (18.7)	34 (6.1)	14.760 (<0.001)
	men	2.37 (1.18)	155 (30.7)	122 (24.2)	140 (27.7)	64 (12.7)	24 (4.8)	
Sleep problems (drowsiness despite adequate sleep intake or insomnia, disturbed sleep rhythm)	women	2.59 (1.24)	133 (24.0)	136 (24.5)	161 (29.0)	76 (13.7)	49 (8.8)	14.068 (<0.01)
	men	2.38 (1.23)	163 (32.3)	120 (23.8)	118 (23.4)	76 (15.0)	28 (5.5)	
Feeling of lack of energy and/or reduced ability to perform daily tasks	women	2.70 (1.18)	104 (18.7)	139 (25.0)	171 (30.9)	100 (18.0)	41 (7.4)	8.770 (0.005)
	men	2.50 (1.15)	118 (23.4)	143 (28.3)	141 (27.9)	80 (15.8)	23 (4.6)	
Emotional strain and/or mental discomfort	women	2.62 (1.15)	110 (19.8)	144 (26.0)	183 (32.9)	83 (15.0)	35 (6.3)	5.423 (0.215)
	men	2.53 (1.19)	129 (25.5)	115 (22.8)	155 (30.7)	76 (15.0)	30 (6.0)	
Difficulty in focusing attention and organising one’s own mental activity	women	2.32 (1.08)	163 (29.4)	140 (25.2)	184 (33.2)	50 (9.0)	18 (3.2)	2.755 (0.767)
	men	2.30 (1.14)	162 (32.1)	124 (24.6)	148 (29.3)	50 (9.8)	21 (4.2)	

**Table 4 ijerph-20-05407-t004:** Individual items of Concerns Checklist (M, SD, and frequencies) by gender (n = 1060; n (%)).

Concerns	Gender	M (SD)	Not at All	Minor Level	Moderately	High Level	Very High Level	Chi^2^ (*p*)
Losing someone you love	women	3.14 (1.21)	59 (10.6)	101 (18.2)	195 (35.1)	105 (18.9)	95 (17.1)	12.012 (<0.05)
	men	2.93 (1.16)	68 (13.5)	102 (20.2)	184 (36.4)	100 (19.8)	51 (10.1)	
Health system overload	women	3.63 (1.08)	27 (4.9)	51 (9.2)	149 (26.8)	204 (36.8)	124 (22.3)	18.820 (<0.001)
	men	3.35 (1.14)	46 (9.1)	51 (10.1)	170 (33.7)	155 (30.7)	83 (16.4)	
Unemployment growth	women	3.35 (1.07)	38 (6.8)	66 (11.9)	189 (34.1)	190 (34.2)	72 (13.0)	21.373 (<0.001)
	men	3.05 (1.14)	66 (13.1)	69 (13.7)	192 (38.0)	128 (25.3)	50 (9.9)	
Price increase	women	4.28 (0.92)	9 (1.6)	17 (3.1)	72 (13.0)	171 (30.8)	286 (51.5)	15.885 (<0.01)
	men	4.04 (1.09)	20 (4.0)	26 (5.1)	91 (18.0)	145 (28.7)	223 (44.2)	
Financial difficulties due to loss of income	women	3.48 (1.15)	33 (5.9)	74 (13.3)	168 (30.3)	156 (28.1)	124 (22.3)	16.856 (<0.01)
	men	3.20 (1.18)	51 (10.1)	74 (14.7)	182 (36.0)	118 (23.4)	80 (15.8)	
The occurrence of an economic crisis	women	3.87 (0.99)	12 (2.2)	39 (7.0)	123 (22.2)	218 (39.3)	163 (29.4)	30.860 (0.001)
	men	3.51 (1.10)	27 (5.3)	58 (11.5)	151 (29.9)	167 (33.1)	102 (20.2)	
Re-Lockdown	women	3.61 (1.06)	28 (5.0)	45 (8.1)	159 (28.6)	205 (36.9)	118 (21.3)	22.228 (<0.001)
	men	3.32 (1.17)	44 (8.7)	69 (13.7)	162 (32.1)	141 (27.9)	89 (17.6)	
Unpredictability and uncertainty of the course of the pandemic	women	3.66 (1.01)	22 (4.0)	31 (5.6)	182 (32.8)	198 (35.7)	122 (22.0)	23.200 (<0.001)
	men	3.36 (1.11)	39 (7.7)	55 (10.9)	175 (34.7)	157 (31.1)	79 (15.6)	
Disease other than COVID-19 in a pandemic situation	women	3.33 (1.04)	30 (5.4)	71 (12.8)	218 (39.3)	157 (28.3)	79 (14.2)	8.530 (<0.01)
	men	3.15 (1.07)	42 (8.3)	71 (14.0)	216 (42.8)	120 (23.8)	56 (11.1)	
I can be a source of infection for my loved ones	women	2.63 (1.02)	78 (14.0)	174 (31.4)	204 (36.8)	75 (13.5)	24 (4.3)	6.155 (0.693)
	men	2.60 (1.04)	89 (17.6)	129 (25.5)	199 (39.4)	70 (13.9)	18 (3.6)	

## Data Availability

The data analyzed during the current study are available.

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
