# Peer review of "Gender-Specificity of Fatigue and Concerns Related to the COVID-19 Pandemic—A Report on the Polish Population"

_ijerph, 2023, doi:10.3390/ijerph20075407_

Round 1
Reviewer 1 Report
I consider the research is relevant and up-to-date, and the article is well written and well organized. The goals are clearly presented and the methods are appropriate. I do not identify any major issue, but I recommend some improvements:
In the Abstract, when the authors present results: “The analysis showed differences in index values between men and women in terms of overall PFS, information PFS, SFS and pandemic-related concerns”, the direction of the differences should be clearly stated, as “Women have more…”.
Line 163, Statistical analysis: The ANOVA was used for several (and not one only) dependent variables; when this happens, Bonferroni correction should be used in order to counteract the multiple comparisons problem. The subsequent changes in the Results should be made.
Line 177, Subjects: Have the authors asked participants about the existence of psychological symptoms and/or psychiatric disorders before the Covid pandemic? It should have been assessed and if they don´t this must be added to the limitations; this assessment would allow the answer to a question in line with the one the authors state in line 307: “This raises the question of to what extent the pandemic generates these symptoms and to what extent it is an exacerbating factor for already existing problems in women (e.g. sleep problems and lack of energy to carry out daily tasks).”
Line 336: “On the one hand, higher levels of anxiety are conducive to coping with stress focused on emotion rather than the problem”- a reference should be here.
Line 356: “According to the study, young adults, compared to older age groups, reported the highest levels of stress, the least ability to cope with this stress and the greatest difficulty in making daily and important decisions [29]. They were more likely to report such symptoms as headaches, fatigue and sleep issues [29,30]”- these sentences are unclear, as the authors state “according to the study” (to the present study) but then have references at the end of the sentence. If they want to state that their results are in line with the one of other studies…they should reformulate.
Author Response
Point 1: In the Abstract, when the authors present results: “The analysis showed differences in index values between men and women in terms of overall PFS, information PFS, SFS and pandemic-related concerns”, the direction of the differences should be clearly stated, as “Women have more…”.
Response 1: Thank you for this remark. Suggested changes to the abstract have been made.
Point 2: Line 163, Statistical analysis: The ANOVA was used for several (and not one only) dependent variables; when this happens, Bonferroni correction should be used in order to counteract the multiple comparisons problem. The subsequent changes in the Results should be made.
Response 2:
If test F was significant we conducted multiple comparisons using Scheffe test (more conservative equivalent of Bonferroni test, used if the assumption of homogeneity of variance was fulfilled, Levene test, p>0.05) or Tamhane test (if Levene test, p<0.05). These analyses refer to the results presented in Supplementary material (tables S5, S6, S7, S8). We presented the results of multiple comparisons only descriptively, but based on post hoc tests. To increase precision of description, we added the following phrase in a bracket: with Scheffe or Tamhane tests for correction in multiple comparisons, depending on homogeneity of variance in the Statistical analyses section.
Point 3: Line 177, Subjects: Have the authors asked participants about the existence of psychological symptoms and/or psychiatric disorders before the Covid pandemic? It should have been assessed and if they don´t this must be added to the limitations; this assessment would allow the answer to a question in line with the one the authors state in line 307: “This raises the question of to what extent the pandemic generates these symptoms and to what extent it is an exacerbating factor for already existing problems in women (e.g. sleep problems and lack of energy to carry out daily tasks).”
Response 3: Information regarding the lack of assessment of mental problems before the pandemic was added to the study's limitations.
Point 4: Line 336: “On the one hand, higher levels of anxiety are conducive to coping with stress focused on emotion rather than the problem”- a reference should be here.
Response 4: Reference added.
Point 5: Line 356: “According to the study, young adults, compared to older age groups, reported the highest levels of stress, the least ability to cope with this stress and the greatest difficulty in making daily and important decisions [29]. They were more likely to report such symptoms as headaches, fatigue and sleep issues [29,30]”- these sentences are unclear, as the authors state “according to the study” (to the present study) but then have references at the end of the sentence. If they want to state that their results are in line with the one of other studies…they should reformulate.
Response 5: A section of the text has been reworded to improve clarity.
Reviewer 2 Report
Row Comment
21 Instead of “sociodemographic survey” use “sociodemographic questions”.
27-31 Conclusion should be focused on the effect of gender (with respect to the aim of the survey and title of the submission).
48 Approaches rather than concepts.
59-72 The rationale for the whole paragraph is unclear. Is it really necessary?
77-79 Relevant evidence for such a statement should be provided.
84 Please, specify the “economic consequences. For whom?
85-86 Is it a hypothesis, a statement, or a finding?
92 Explain in detail why women are more likely to experience anxiety? Is there any added value of your research, or do you just replicate such findings?
100-101 The sentence shall fit better, if it is placed at the end of row 72; consider that, please.
110 Sampling procedure must be significantly improved and the details must be added. What particular sampling technique was used? The description resembles proportionate stratified random sampling, but is it not obvious. How is the theoretical population defined? What sampling frame was used? Was RDD (random digit dialing) used during CATI? What was the response rate? Comparison of the sample with theoretical population should be provided. Was any weighting procedure used?
115 Was that a plan, or is it a real outcome of the field-works?
139 The term “parameter” is associated with the population, not the sample.
173 The name of the coefficient should be used consistently throughout the whole paper (compare with e.g., row 156).
178-189 The whole paragraph brings low added value because it only repeats what is in the table. May the text could explain to which extent the sample is representative in comparison to the theoretical population.
179 Phrase “a slight predominance of women” lacks statistical rigor and should be avoided in scientific journals.
188 The value 65.8 % seems incorrect (the table shows 64.8%).
Table 1 Value labels for “chronic diseases” and “COVID-19 vaccination” are confusing.
Absolute values and percentages should be switched (absolute frequency is less important and therefore it should be just in brackets).
“Woman” and “Man” should be replaced either by “Women” and “Men”, or even better “Female(s)” and “Males(s)”.
Table 2, 3, 4 Instead of the cross-tabs, the distribution of the composite index should be presented; it is declared that the statement build-up a scale (or a common continuum), therefore it is desirable to see the distribution of that: apart from mean (median) and SD also minimum, maximum, skewness, kurtosis, possible floor and ceiling effects, and the correlation matrix. After that the subsegment comparisons could be presented. Would be fine to perform ANOVA post-hoc to identify different subsegments.
Author Response
Point 1: Row 21: Instead of “sociodemographic survey” use “sociodemographic questions”.
Response 1: Amended in the text as indicated.
Point 2: Row 27-31: Conclusion should be focused on the effect of gender (with respect to the aim of the survey and title of the submission).
Response 2: Reformulated the conclusion to focus on gender differences.
Point 3: Row 48: Approaches rather than concepts.
Response 3: : Amended in the text as indicated.
Point 4: Row 59-72:The rationale for the whole paragraph is unclear. Is it really necessary?
Response 4: Rudroff's concept presented in the paragraph refers to pandemic fatigue in the context of symptoms of reduced psycho-physical performance as a result of the COVID-19 pandemic. It differs from the WHO definition, which focuses on pandemic fatigue mainly in the context of adherence to health recommendations. Rudrof's concept is relevant in the context of the research tools used and the factors analyzed, therefore the authors suggest leaving this section.
Point 5: Row 77-79:Relevant evidence for such a statement should be provided.
Response 5: References added
Point 6: Row 84: Please, specify the “economic consequences. For whom?
Response 6: Supplemented in the text
Point 7: Row 85-86: Is it a hypothesis, a statement, or a finding?
Response 7: This is a statement based on an analysis of the results of other studies. A reference has been added to dispel doubts about this sentence.
Point 8: Row 92: Explain in detail why women are more likely to experience anxiety? Is there any added value of your research, or do you just replicate such findings?
Response 8: The study of anxiety and concerns was one component of our study. The study confirmed that women are more concerned about COVID-19. In addition, we have defined the areas of these concerns.
Point 9: Row 100-101:The sentence shall fit better, if it is placed at the end of row 72; consider that, please.
Response 9: Sentence moved to the indicated place.
Point 10: Row 110: Sampling procedure must be significantly improved and the details must be added. What particular sampling technique was used? The description resembles proportionate stratified random sampling, but is it not obvious. How is the theoretical population defined? What sampling frame was used? Was RDD (random digit dialing) used during CATI? What was the response rate? Comparison of the sample with theoretical population should be provided. Was any weighting procedure used?
Response 10: The methodology section was supplemented by the method of obtaining the study sample (e.g., random sampling oparat). No weighing procedure was used.
Point 11: Row 115: Was that a plan, or is it a real outcome of the field-works?
Response 11: Maintaining the age and gender ratio in each voivodeship was an actual goal that was realized.
Point 12: Row 139:The term “parameter” is associated with the population, not the sample.
Response 12: "Parameter" has been changed to "element" as more appropriate in this context.
Point 13: Row 173:The name of the coefficient should be used consistently throughout the whole paper (compare with e.g., row 156).
Response 13:
The whole manuscript was checked and the name of the coefficient has been unified.
Point 14: Row 178-189:The whole paragraph brings low added value because it only repeats what is in the table. May the text could explain to which extent the sample is representative in comparison to the theoretical population.
Response 14: the study was conducted on a representative sample for the inhabitants of Poland
Point 15: Row 179: Phrase “a slight predominance of women” lacks statistical rigor and should be avoided in scientific journals.
Response 15: Removed the mentioned part of the sentence
Point 16: Row 188: The value 65.8 % seems incorrect (the table shows 64.8%).
Response 16: Changed the value in the text to 64.8%
Point 17: a)Table 1: Value labels for “chronic diseases” and “COVID-19 vaccination” are confusing.
- b) Absolute values and percentages should be switched (absolute frequency is less important and therefore it should be just in brackets).
- c) “Woman” and “Man” should be replaced either by “Women” and “Men”, or even better “Female(s)” and “Males(s)”.
Response 17:
- a) values labels have been reworded
b)Notation: Absolute values and percentages have been switched.
c)Column names replaced with „Females” and „Males”
Point 18: Table 2, 3, 4 Instead of the cross-tabs, the distribution of the composite index should be presented; it is declared that the statement build-up a scale (or a common continuum), therefore it is desirable to see the distribution of that: apart from mean (median) and SD also minimum, maximum, skewness, kurtosis, possible floor and ceiling effects, and the correlation matrix. After that the subsegment comparisons could be presented. Would be fine to perform ANOVA post-hoc to identify different subsegments.
Response 18:
Due to the fact that the tools used in this study and data of this type have not been published before, in this section we wanted to show cross-gender comparisons on all specific indicators/symptoms. Examining the relationship between indicators was not the purpose of this study, therefore we did not present correlation data. The suggestion to show more detailed data on aggregate indicators seems to us very valuable, so we will definitely use this advice in the future.
Round 2
Reviewer 2 Report
Thank you for accepting the proposed improvements. I have no further queries and consider the revised version of the article acceptable for publication.
Author Response
Thank you very much for a thorough analysis of our revised manuscript and very valuable comments.